# Association between (ΔPaO$_2$/FiO$_2$)/PEEP and in-hospital mortality in patients with COVID-19 pneumonia: A secondary analysis

**Youli Chen, Huangen Li, Jinhuang Lin, Zhiwei Su, Tianlai Lin**◉*

Intensive Care Unit, Fujian Medical University Affiliated First Quanzhou Hospital, Quanzhou, Fujian, PR China

* ltl688@163.com

## Abstract

### Background

The arterial pressure of oxygen (PaO$_2$)/inspiratory fraction of oxygen (FiO$_2$) is associated with in-hospital mortality in patients with Coronavirus Disease 2019 (COVID-19) pneumonia. ΔPaO$_2$/FiO$_2$ [the difference between PaO$_2$/FiO$_2$ after 24 h of invasive mechanical ventilation (IMV) and PaO$_2$/FiO$_2$ before IMV] is associated with in-hospital mortality. However, the value of PaO$_2$ can be influenced by the end-expiratory pressure (PEEP). To the best of our knowledge, the relationship between the ratio of (ΔPaO$_2$/FiO$_2$)/PEEP and in-hospital mortality remains unclear. This study aimed to evaluate their association.

### Methods

The study was conducted in southern Peru from April 2020 to April 2021. A total of 200 patients with COVID-19 pneumonia requiring IMV were included in the present study. We analyzed the association between (ΔPaO$_2$/FiO$_2$)/PEEP and in-hospital mortality by Cox proportional hazards regression models.

### Results

The median (ΔPaO$_2$/FiO$_2$)/PEEP was 11.78 mmHg/cmH$_2$O [interquartile range (IQR) 8.79–16.08 mmHg/cmH$_2$O], with a range of 1 to 44.36 mmHg/cmH$_2$O. Patients were divided equally into two groups [low group (< 11.80 mmHg/cmH$_2$O), and high group ($\geq$ 11.80 mmHg/cmH$_2$O)] according to the (ΔPaO$_2$/FiO$_2$)/PEEP ratio. In-hospital mortality was lower in the high (ΔPaO$_2$/FiO$_2$)/PEEP group than in the low (ΔPaO$_2$/FiO$_2$)/PEEP group [18 (13%) vs. 38 (38%)]; hazard ratio (HR), 0.33 [95% confidence intervals (CI), 0.17–0.61, P < 0.001], adjusted HR, 0.32 (95% CI, 0.11–0.94, P = 0.038). The finding that the high (ΔPaO$_2$/FiO$_2$)/PEEP group exhibited a lower risk of in-hospital mortality compared to the low (ΔPaO$_2$/FiO$_2$)/PEEP group was consistent with the results from the sensitivity analysis. After adjusting for confounding variables, we found that each unit increase in (ΔPaO$_2$/FiO$_2$)/PEEP was associated with a 12% reduction in the risk of in-hospital mortality (HR, 0.88, 95%CI, 0.80–0.97, P = 0.013).

**Data Availability Statement:** Data were freely extracted from (https://www.ncbi.nlm.nih.gov/

pmc/articles/PMC9756861/bin/peerj-10-14290-s004.xls).

**Funding:** The author(s) received no specific funding for this work.

**Competing interests:** The authors have declared that no competing interests exist.

## Conclusions

The $(\Delta PaO_2/FiO_2)/PEEP$ ratio was associated with in-hospital mortality in patients with COVID-19 pneumonia. $(\Delta PaO_2/FiO_2)/PEEP$ might be a marker of disease severity in COVID-19 patients.

## Introduction

Respiratory failure is a common complication in patients with severe coronavirus disease 2019 (COVID-19). It is a common reason for the need for invasive mechanical ventilation (IMV) and a significant contributor to mortality among individuals with COVID-19 [1]. Studies reported that a proportion of COVID-19 patients (23.6% to 33.1%) required IMV [2,3]. In addition, the mortality rate of COVID-19 patients requiring IMV was approximately 28.6% to 60.4% [3,4]. The arterial pressure of oxygen ($PaO_2$)/inspiratory fraction of oxygen ($FiO_2$) was both an important indicator of tracheal intubation and a correlate of mortality. Previous studies focused on the relationship between $PaO_2/FiO_2$ and prognosis at the time of admission to the hospital or intensive care unit (ICU) [5,6]. Their studies showed that lower $PaO_2/FiO_2$ was associated with increased in-hospital mortality in COVID-19 patients [7,8]. However, a study by Arnold-Day C et al. [9] showed that $PaO_2/FiO_2$ was not an independent risk factor for death in patients requiring IMV. Another study by Miguel et al. [10] showed that there was no association between $PaO_2/FiO_2$ before IMV and mortality. Therefore, $PaO_2/FiO_2$ might not be a good assessment of prognosis in COVID-19 patients requiring IMV. Miguel et al. also found that the difference between $PaO_2/FiO_2$ after 24 h of IMV and $PaO_2/FiO_2$ before IMV ($\Delta PaO_2/FiO_2$) was associated with in-hospital mortality. However, the value of $PaO_2$ could be influenced by end-expiratory pressure (PEEP), and patients with higher PEEP might have more severe lung injury than those with lower PEEP for the same $\Delta PaO_2/FiO_2$ [11]. Palanidurai S et al. [11] created the $(PaO_2/FiO_2)/PEEP$ ratio and found that $(PaO_2/FiO_2)/PEEP$ ratio was associated with mortality in patients with acute respiratory distress syndrome. In this study, we aimed to evaluate the relationship between $(\Delta PaO_2/FiO_2)/PEEP$ and in-hospital mortality in patients with COVID-19 pneumonia.

## Materials and methods

### Study design and participants

This study assessed a retrospective single-center cohort study conducted by Miguel et al [10] from April 2020 to April 2021 in southern Peru. The study reanalyzed 200 patients with COVID-19 pneumonia requiring IMV treatment. This was a secondary analysis study. Mechanical ventilation management strategies and prone position ventilation management strategies for this cohort were described in full previously [10]. The study was approved by the Ethics Committee of Faculty of Health Sciences of the Private University of Tacna (identification code: N391-2021-UPT/FACSA-D). As this was an observational study, and the data were anonymized, the requirement for informed consent was waived. Data were freely extracted from Miguel et al. [10] (https://www.ncbi.nlm.nih.gov/pmc/articles/PMC9756861/bin/peerj-10-14290-s004.xls).

### Stratification definitions

High white blood cells: $> 10 \times 10^9$/L. High lymphocyte: $> 1 \times 10^9$/L. High platelets: $> 100 \times 10^9$/L. High procalcitonin: $> 0.5$ ng/ml. High C-reactive protein: $> 100$ mg/L. High

alanine aminotransferase: $> 40$ U/L. High aspartate aminotransferase: $> 40$ U/L. High creatinine phosphokinase: $> 200$ U/L. High creatinine phosphokinase-MB: $> 25$ U/L [12].

High sequential organ failure assessment score: $> 4$ [13].

High plateau pressure after 24 h of IMV: $> 30$ cmH$_2$O. High driving pressure after 24 h of IMV: $> 15$ cmH$_2$O [10].

($\Delta$PaO$_2$/FiO$_2$)/PEEP: the difference between PaO$_2$/FiO$_2$ after 24 h of IMV and PaO$_2$/FiO$_2$ before IMV)/PEEP.

## Statistical analysis

Continuous data were presented as mean ± standard deviation (SD) if normally distributed, and median [interquartile range (IQR)], if data were non-normal. Categorical variables were presented as frequency and percentages (n; %). The patients were divided equally into two groups [low group ($< 11.80$ mmHg/cmH$_2$O), and high group ($\geq 11.80$ mmHg/cmH$_2$O)] according to ($\Delta$PaO$_2$/FiO$_2$)/PEEP. The Kruskal-Wallis test was used for the comparison of continuous variables. Comparisons of categorical variables were performed using the chi-square test or Fisher's exact probability test. To explore the relationship between ($\Delta$PaO$_2$/ FiO$_2$)/PEEP and in-hospital mortality, we performed univariate and multivariate analyses based on the Cox proportional hazards model. In multivariate analysis, we present results for both unadjusted and fully adjusted analytical models. To maximize statistical power and remove bias, we used multiple imputation (MI). To compare in-hospital mortality in the different groups, subgroup analyses were performed using a stratified Cox proportional hazards model. The effects of ($\Delta$PaO$_2$/FiO$_2$)/PEEP on in-hospital mortality were assessed using Kaplan-Meier curves (log-rank test). We applied a smooth curve technique to estimate the shape between ($\Delta$PaO$_2$/FiO$_2$)/PEEP and in-hospital mortality by restricted cubic spline regression. Data were analyzed using EmpowerStats (www.empowerstats.com, X&Y solutions, Boston, Massachusetts, USA) and R software version 3.6.1 (http://www.r-project.org). A two-tailed p-value of $< 0.05$ was considered statistically significant.

## Results

### Patient and baseline characteristics

Demographic and clinical characteristics of patients were shown in Table 1 according to ($\Delta$PaO$_2$/FiO$_2$)/PEEP. The mean age of patients was 54.29 ± 12.20 years. Among 200 patients, 42 (21%) were female and 158 (79%) were male. There was a statistical difference in comorbidities between the two groups, with chronic kidney disease and heart failure being more common comorbidities in the low ($\Delta$PaO$_2$/FiO$_2$)/PEEP group (chronic kidney disease, 8% vs. 0, P = 0.007; heart failure, 10% vs. 0, P = 0.001). Lung damage on computed tomography and platform pressure levels were higher in the low ($\Delta$PaO$_2$/FiO$_2$)/PEEP group (lung damage on computed tomography $> 50$%, 73% vs. 52%, P = 0.002; platform pressure levels $> 30$ cmH$_2$O, 33% vs. 20%, P = 0.037). Sepsis, septic shock, acute kidney failure, and renal replacement therapy were also more common in the low ($\Delta$PaO$_2$/FiO$_2$)/PEEP group (sepsis, 80% vs 63%, P = 0.008; septic shock, 37% vs. 21%, P = 0.013; acute kidney failure, 22% vs. 7%, P = 0.003, renal replacement therapy, 12% vs. 3%, P = 0.016). We found that PaO$_2$/FiO$_2$ after 24 h of IMV were lower and PEEP was higher in the low ($\Delta$PaO$_2$/FiO$_2$)/PEEP group compared with the high ($\Delta$PaO$_2$/FiO$_2$)/PEEP group (197.85 ± 55 vs. 301.49 ± 71.75, P $< 0.001$). There was no difference in PaO$_2$/FiO$_2$ before entering IMV between the two groups (100.65 ± 43.62 vs. 99.81 ± 34.18, P = 0.880).

Data on the immunosuppression, sequential organ failure assessment, corticosteroids, and arrhythmia were missing for 1 patient, on the C-reactive protein for 10 patients, on the

**Table 1. Demographic and clinical characteristics of patients according to (ΔPaO$_2$/FiO$_2$)/PEEP.**

| Variable | Low (< 11.80) | High (≥ 11.80) | χ$^2$ | p-Value |
|---|---|---|---|---|
| Number | 100 | 100 | | |
| Female | 20 (20%) | 22 (22%) | 0.121 | 0.728 |
| Age (years) | | | 0.030 | 0.861 |
| < 65 | 79 (79%) | 80 (80%) | | |
| ≥ 65 | 21 (21%) | 20 (20%) | | |
| Obesity | 61 (61%) | 57 (57%) | 0.331 | 0.565 |
| Hypertension | 31 (31%) | 22 (22%) | 2.079 | 0.149 |
| Diabetes | 22 (22%) | 21 (21%) | 0.030 | 0.863 |
| Chronic renal insufficiency | 8 (8%) | 0 (0) | - | 0.007 |
| Heart failure | 10 (10%) | 0 (0) | 10.526 | 0.001 |
| Asthma | 12 (12%) | 18 (18%) | 1.142 | 0.235 |
| Immunosuppression | 11 (11%) | 7 (7.07%) | 0.934 | 0.334 |
| White blood cells (×10$^9$/L) | | | 0.731 | 0.393 |
| ≤ 10 | 53 (53%) | 59 (59%) | | |
| > 10 | 47 (47%) | 41 (41%) | | |
| Lymphocytes (×10$^9$/L) | | | 0.026 | 0.873 |
| ≤ 1 | 73 (73%) | 74 (74%) | | |
| > 1 | 27 (27%) | 26 (26%) | | |
| Platelets (×10$^9$/L) | | | 2.922 | 0.087 |
| ≤ 300 | 62 (62%) | 50 (50%) | | |
| > 300 | 38 (38%) | 50 (50%) | | |
| C-reactive protein (mg/L) | | | 1.027 | 0.311 |
| ≤ 100 | 32 (33.33%) | 38 (40.43%) | | |
| > 100 | 64 (66.67%) | 56 (59.57%) | | |
| Procalcitonin (ng/mL) | | | 3.128 | 0.077 |
| ≤ 0.5 | 55 (75.34%) | 55 (87.30%) | | |
| > 0.5 | 18 (24.66%) | 8 (12.70%) | | |
| Alanine aminotransferase (U/L) | | | 0.226 | 0.635 |
| ≤ 40 | 26 (26%) | 29 (29%) | | |
| > 40 | 74 (74%) | 71 (71%) | | |
| Aspartate aminotransferase (U/L) | | | 0.546 | 0.460 |
| ≤ 40 | 33 (33%) | 38 (38%) | | |
| > 40 | 67 (67%) | 62 (62%) | | |
| Creatinine phosphokinase-Total (U/L) | | | 1.602 | 0.206 |
| ≤ 200 | 59 (62.77%) | 63 (71.59%) | | |
| > 200 | 35 (37.23%) | 25 (28.41%) | | |
| Creatinine phosphokinase-MB (U/L) | | | 0.078 | 0.780 |
| ≤ 25 | 45 (46.88%) | 45 (48.91%) | | |
| > 25 | 51 (53.12%) | 47 (51.09%) | | |
| Lung damage on computed tomography | | | 9.408 | 0.002 |
| ≤ 50% | 27 (27%) | 48 (48%) | | |
| > 50% | 73 (73%) | 52 (52%) | | |
| Sequential organ failure assessment | | | 1.962 | 0.161 |
| ≤ 4 | 61 (61.62%) | 71 (71%) | | |
| > 4 | 38 (38.38%) | 29 (29%) | | |
| Antibiotics | 99 (99%) | 98 (98%) | - | 1 |
| Corticosteroids | 98 (98%) | 96 (96.97%) | - | 0.683 |

*(Continued)*

**Table 1.** (Continued)

| Variable | Low (< 11.80) | High (≥ 11.80) | $\chi^2$ | p-Value |
|---|---|---|---|---|
| Colchicine | 32 (32%) | 20 (20%) | 3.742 | 0.053 |
| Tocilizumab | 14 (14%) | 12 (12%) | 0.177 | 0.674 |
| Renal replacement therapy | 12 (12%) | 3 (3%) | 5.834 | 0.016 |
| Sepsis | 80 (80%) | 63 (63%) | 7.091 | 0.008 |
| Septic shock | 37 (37%) | 21 (21%) | 6.217 | 0.013 |
| Acute kidney failure | 22 (22%) | 7 (7%) | 9.074 | 0.003 |
| Arrhythmia | 9 (9.09%) | 4 (4%) | 2.112 | 0.146 |
| Pneumonia associated with IMV | 26 (26%) | 18 (18%) | 1.865 | 0.172 |
| Catheter-associated bacteremia | 6 (6%) | 5 (5%) | 0.096 | 0.756 |
| Plateau pressure 24 h after IMV (cmH$_2$O) | | | 4.338 | 0.037 |
| $\leq$ 30 | 67 (67%) | 80 (80%) | | |
| > 30 | 33 (33%) | 20 (20%) | | |
| Driving pressure 24 h after IMV (cmH$_2$O) | | | 1.025 | 0.311 |
| $\leq$ 15 | 36 (36%) | 43 (43%) | | |
| > 15 | 64 (64%) | 57 (57%) | | |
| Tidal volume 24 h after IMV (ml) | 459.47 ± 82.28 | 474.27 ± 72.53 | t = -1.349 | 0.179 |
| PaO$_2$/FiO$_2$ before IMV (mmHg) | 100.65 ± 43.62 | 99.81 ± 34.18 | t = 0.151 | 0.880 |
| PaO$_2$/FiO$_2$ 24 h after IMV (mmHg) | 197.85 ± 55 | 301.49 ± 71.75 | t = -11.464 | <0.001 |
| PEEP 24 h after IMV (cmH$_2$O) | 12.70 ± 2 | 11.69 ± 1.65 | t = 3.893 | <0.001 |
| IMV LOS (days) | 11 (7–19) | 8.50 (5–13.25) | z = -2.297 | 0.023 |
| ICU LOS (days) | 11 (7–17.25) | 9 (5–14) | z = -1.886 | 0.061 |
| Hospital LOS (days) | 20 (15–28) | 20 (13–29) | z = -0.223 | 0.822 |
| In-hospital mortality | 38 (38%) | 13 (13%) | 16.450 | <0.001 |

IMV, invasive mechanical ventilation; PEEP, end-expiratory pressure; ICU, intensive care unit; LOS, length of stay.

procalcitonin for 64 patients, on the creatinine phosphokinase-Total for 18 patients, on the creatinine phosphokinase-MB for 12 patients.

The duration of mechanical ventilation was longer in the low ($\Delta PaO_2/FiO_2$)/PEEP group compared with the high ($\Delta PaO_2/FiO_2$)/PEEP group [11 (7–19) vs. 8.50 (5–13.25)]. Mortality was higher in the low ($\Delta PaO_2/FiO_2$)/PEEP group compared to the high ($\Delta PaO_2/FiO_2$)/PEEP group [38 (38%) vs. 13 (13%)] (Table 1).

## Univariate analysis of mortality

As reported in Table 2, in the univariate analysis, we found that diabetes, chronic renal insufficiency, heart failure, immunosuppression, lung damage on computed tomography > 50%, sequential organ failure assessment score > 4, tocilizumab, renal replacement therapy, septic shock, acute kidney failure, pneumonia associated with IMV, and plateau pressure 24 h after IMV were associated with higher in-hospital death risk. Based on clinical and statistical reasons (S1 Table), we chose confounding factors including sex, age, obesity, diabetes, chronic renal insufficiency, heart failure, asthma, immunosuppression, platelets, C-reactive protein, procalcitonin, alanine aminotransferase, aspartate aminotransferase, lung damage on computed tomography, sequential organ failure assessment score, tocilizumab, renal replacement therapy, sepsis, septic shock, acute kidney failure, pneumonia associated with IMV, plateau pressure 24 h after IMV, and driving pressure 24 h after IMV.

**Table 2. Univariate analysis of prognostic factors.**

| Variable | In-hospital mortality HR (95% CI) | P-value |
|---|---|---|
| Sex | | |
| Female | 1.0 | |
| Male | 0.90 (0.45, 1.81) | 0.770 |
| Age (year) | | |
| < 65 | 1.0 | |
| ≥ 65 | 1.35 (0.76, 2.41) | 0.311 |
| Obesity | | |
| No | 1.0 | |
| Yes | 1.16 (0.66, 2.03) | 0.605 |
| Hypertension | | |
| No | 1.0 | |
| Yes | 1.21 (0.68, 2.15) | 0.525 |
| Diabetes | | |
| No | 1.0 | |
| Yes | 1.91 (1.04, 3.49) | 0.036 |
| Chronic renal insufficiency | | |
| No | 1.0 | |
| Yes | 6.86 (3.06, 15.38) | <0.001 |
| Heart failure | | |
| No | 1.0 | |
| Yes | 2.85 (1.32, 6.12) | 0.007 |
| Asthma | | |
| No | 1.0 | |
| Yes | 1.27 (0.66, 2.44) | 0.469 |
| Immunosuppression | | |
| No | 1.0 | |
| Yes | 3.54 (1.80, 6.96) | <0.001 |
| White blood cells (×10$^9$/L) | | |
| ≤ 10 | 1.0 | |
| > 10 | 1.46 (0.83, 2.56) | 0.184 |
| Lymphocytes (×10$^9$/L) | | |
| ≤ 1 | 1.0 | |
| > 1 | 1.39 (0.75, 2.59) | 0.296 |
| Platelets (×10$^9$/L) | | |
| ≤ 300 | 1.0 | |
| > 300 | 0.57 (0.31, 1.03) | 0.064 |
| C-reactive protein (mg/L) | | |
| ≤ 100 | 1.0 | |
| > 100 | 1.87 (0.95, 3.67) | 0.068 |
| Procalcitonin (ng/mL) | | |
| ≤ 0.5 | 1.0 | |
| > 0.5 | 1.26 (0.61, 2.58) | 0.535 |
| Alanine aminotransferase (U/L) | | |
| ≤ 40 | 1.0 | |
| > 40 | 0.63 (0.35, 1.12) | 0.114 |
| Aspartate aminotransferase (U/L) | | |
| ≤ 40 | 1.0 | |

(*Continued*)

**Table 2.** (Continued)

| Variable | In-hospital mortality HR (95% CI) | P-value |
|---|---|---|
| > 40 | 1.01 (0.56, 1.80) | 0.986 |
| Creatinine phosphokinase-Total (U/L) | | |
| ≤ 200 | 1.0 | |
| > 200 | 1.61 (0.91, 2.84) | 0.104 |
| Creatinine phosphokinase-MB (U/L) | | |
| ≤ 25 | 1.0 | |
| > 25 | 1.11 (0.63, 1.97) | 0.712 |
| Lung damage on computed tomography | | |
| ≤ 50% | 1.0 | |
| > 50% | 2.89 (1.23, 6.82) | 0.015 |
| Sequential organ failure assessment | | |
| ≤ 4 | 1.0 | |
| > 4 | 3.20 (1.78, 5.76) | <0.001 |
| Corticosteroids | | |
| No | 1.0 | |
| Yes | 0.91 (0.12, 6.61) | 0.924 |
| Colchicine | | |
| No | 1.0 | |
| Yes | 1.33 (0.75, 2.36) | 0.335 |
| Tocilizumab | | |
| No | 1.0 | |
| Yes | 2.53 (1.31, 4.91) | 0.006 |
| Renal replacement therapy | | |
| No | 1.0 | |
| Yes | 3.75 (1.98, 7.14) | <0.001 |
| Sepsis | | |
| No | 1.0 | |
| Yes | 1.55 (0.69, 3.47) | 0.286 |
| Septic shock | | |
| No | 1.0 | |
| Yes | 2.91 (1.60, 5.29) | <0.001 |
| Acute kidney failure | | |
| No | 1.0 | |
| Yes | 3.25 (1.85, 5.71) | <0.001 |
| Arrhythmia | | |
| No | 1.0 | |
| Yes | 1.79 (0.85, 3.75) | 0.122 |
| Pneumonia associated with IMV | | |
| No | 1.0 | |
| Yes | 3.89 (2.21, 6.84) | <0.001 |
| Catheter-associated bacteremia | | |
| No | 1.0 | |
| Yes | 1.40 (0.55, 3.52) | 0.481 |
| Plateau pressure 24 h after IMV (cmH$_2$O) | | |
| ≤ 30 | 1.0 | |
| > 30 | 2.23 (1.28, 3.88) | 0.004 |
| Driving pressure 24 h after IMV (cmH$_2$O) | | |

(*Continued*)

**Table 2.** (Continued)

| Variable | In-hospital mortality HR (95% CI) | P-value |
|---|---|---|
| ≤ 15 | 1.0 | |
| > 15 | 1.46 (0.81, 2.64) | 0.210 |

HR, hazard ratio; CI, confidence interval; IMV, invasive mechanical ventilation.

## Sensitivity analyses

We conducted a stratified analysis according to baseline characteristics. In subgroup analyses (Table 3), the high (ΔPaO$_2$/FiO$_2$)/PEEP group was associated with lower in-hospital mortality in most strata compared with the low (ΔPaO$_2$/FiO$_2$)/PEEP group. When the sample size was less than 10, we did not perform stratified analyses. As shown in Table 3 and S2 Table, the interaction test was statistically significant in both age and C-reactive protein groups ($p < 0.05$). In patients aged < 65 years, the risk of death was lower in the high group (HR, 0.18, 95% CI, 0.08–0.45, P = 0.026). The risk of death was higher in the low group among patients with C-reactive protein > 100 mg/L (HR, 3.63, 95%CI, 1.28–10.31, P = 0.007).

## Multivariate analyses of (ΔPaO$_2$/FiO$_2$)/PEEP and in-hospital mortality

Out of 200 patients, 51 (25.50%) died. The Kaplan-Meier curves for survival rate were shown in Fig 1. The low (ΔPaO$_2$/FiO$_2$)/PEEP group with the highest mortality (38%) was the reference group (Table 4). As shown in Table 4, the high (ΔPaO$_2$/FiO$_2$)/PEEP levels group was associated with a 67% risk decrease in death (HR, 0.33, 95%CI, 0.17–0.61, P < 0.001). After adjusting for confounding factors, the relationship was still robust (HR, 0.32, 95%CI, 0.11–0.94, P = 0.038). After adjusting for confounding variables, we found that each unit increase in (ΔPaO$_2$/FiO$_2$)/PEEP was associated with a 12% reduction in the risk of in-hospital mortality (HR, 0.88, 95%CI, 0.80–0.97, P = 0.013). After excluding outliers (values less than Q1–1.5*IQR or greater than Q3 + 1.5*IQR), where IQR is the interquartile range and Q1 and Q3 are the first and third quartiles, respectively, the risk of death decreased with (ΔPaO2/FiO2)/PEEP [(ΔPaO$_2$/FiO$_2$)/PEEP < 27.02 cmH$_2$O] increasing (Fig 2).

   Model I adjusted for: sex, age (< 65, ≥ 65), obesity, diabetes, chronic renal insufficiency, heart failure, asthma, immunosuppression, platelets (≤ 300, > 300), C-reactive protein (≤ 100, > 100), procalcitonin (≤ 0.5, > 0.5), alanine aminotransferase (≤ 40, > 40), aspartate aminotransferase (≤ 40, > 40), lung damage on computed tomography (≤ 50%, > 50%), sequential organ failure assessment (≤ 4, > 4), tocilizumab, renal replacement therapy, sepsis, septic shock, acute kidney failure, pneumonia associated with IMV, plateau pressure 24 h after IMV (≤ 30, > 30), driving pressure 24 h after IMV (≤ 15, > 15).

   Model II adjusted for: sex, age, obesity, diabetes, chronic renal insufficiency, heart failure, asthma, immunosuppression, Platelets, C-reactive protein, procalcitonin, alanine aminotransferase, aspartate aminotransferase, lung damage on computed tomography, sequential organ failure assessment, tocilizumab, renal replacement therapy, sepsis, septic shock, acute kidney failure, pneumonia associated with IMV, plateau pressure 24 h after IMV, driving pressure 24 h after IMV.

   Adjusted for sex, age, obesity, diabetes, chronic renal insufficiency, heart failure, asthma, immunosuppression, platelets, C-reactive protein, procalcitonin, alanine aminotransferase, aspartate aminotransferase, lung damage on computed tomography, sequential organ failure assessment, tocilizumab, renal replacement therapy, sepsis, septic shock, acute kidney failure,

**Table 3. Stratified analysis of $(\Delta PaO_2/FiO_2)/PEEP$ and in-hospital mortality.**

| Variable | Total | Death | HR (95% CI) | | P for interaction |
|---|---|---|---|---|---|
| | | | Low ($< 11.80$) | High ($\geq 11.80$) | |
| Sex | | | | | 0.5523 |
| Female | 42 | 10 | 1.0 | 0.46 (0.12, 1.79) | |
| Male | 158 | 41 | 1.0 | 0.29 (0.14, 0.60) | |
| Age (year) | | | | | 0.0264 |
| $< 65$ | 159 | 33 | 1.0 | 0.20 (0.08, 0.48) | |
| $\geq 65$ | 41 | 18 | 1.0 | 0.85 (0.32, 2.26) | |
| Obesity | | | | | 0.6244 |
| No | 82 | 21 | 1.0 | 0.26 (0.10, 0.68) | |
| Yes | 118 | 30 | 1.0 | 0.37 (0.16, 0.86) | |
| Hypertension | | | | | 0.4759 |
| No | 147 | 33 | 1.0 | 0.38 (0.18, 0.80) | |
| Yes | 53 | 18 | 1.0 | 0.24 (0.07, 0.85) | |
| Diabetes | | | | | 0.4880 |
| No | 157 | 35 | 1.0 | 0.35 (0.17, 0.76) | |
| Yes | 43 | 16 | 1.0 | 0.24 (0.08, 0.77) | |
| Chronic renal insufficiency | | | | | - |
| No | 192 | 44 | 1.0 | 0.38 (0.20, 0.72) | |
| Heart failure | | | | | - |
| No | 190 | 43 | 1.0 | 0.36 (0.19, 0.70) | |
| Asthma | | | | | 0.7396 |
| No | 170 | 38 | 1.0 | 0.33 (0.16, 0.71) | |
| Yes | 30 | 13 | 1.0 | 0.29 (0.09, 0.94) | |
| Immunosuppression | | | | | - |
| No | 181 | 40 | 1.0 | 0.37 (0.19, 0.75) | |
| Yes | 18 | 11 | 1.0 | 0.13 (0.02, 1.05) | |
| White blood cells ($\times 10^9$/L) | | | | | 0.0804 |
| $\leq 10$ | 112 | 22 | 1.0 | 0.57 (0.24, 1.37) | |
| $> 10$ | 88 | 29 | 1.0 | 0.19 (0.07, 0.50) | |
| Lymphocytes ($\times 10^9$/L) | | | | | 0.0550 |
| $\leq 1$ | 147 | 37 | 1.0 | 0.45 (0.22, 0.89) | |
| $> 1$ | 53 | 14 | 1.0 | 0.10 (0.01, 0.75) | |
| Platelets ($\times 10^9$/L) | | | | | 0.1392 |
| $\leq 300$ | 112 | 36 | 1.0 | 0.48 (0.23, 1.01) | |
| $> 300$ | 88 | 15 | 1.0 | 0.16 (0.05, 0.58) | |
| C-reactive protein (mg/L) | | | | | 0.0072 |
| $\leq 100$ | 70 | 11 | 1.0 | 1.37 (0.39, 4.75) | |
| $> 100$ | 120 | 37 | 1.0 | 0.17 (0.07, 0.44) | |
| Procalcitonin (ng/mL) | | | | | 0.5028 |
| $\leq 0.5$ | 110 | 31 | 1.0 | 0.27 (0.12, 0.63) | |
| $> 0.5$ | 26 | 10 | 1.0 | 0.22 (0.03, 1.75) | |
| Alanine aminotransferase (U/L) | | | | | 0.5016 |
| $\leq 40$ | 55 | 18 | 1.0 | 0.45 (0.17, 1.21) | |
| $> 40$ | 145 | 33 | 1.0 | 0.26 (0.11, 0.61) | |
| Aspartate aminotransferase (U/L) | | | | | 0.2649 |
| $\leq 40$ | 71 | 17 | 1.0 | 0.18 (0.05, 0.63) | |
| $> 40$ | 129 | 34 | 1.0 | 0.43 (0.20, 0.89) | |

*(Continued)*

**Table 3.** (*Continued*)

| Variable | Total | Death | HR (95% CI) | | P for interaction |
|---|---|---|---|---|---|
| | | | Low (< 11.80) | High (≥ 11.80) | |
| Creatinine phosphokinase-Total (U/L) | | | | | 0.7565 |
| ≤ 200 | 122 | 29 | 1.0 | 0.30 (0.13, 0.69) | |
| > 200 | 60 | 29 | 1.0 | 0.35 (0.11, 1.11) | |
| Creatinine phosphokinase-MB (U/L) | | | | | 0.5275 |
| ≤ 25 | 90 | 22 | 1.0 | 0.22 (0.08, 0.66) | |
| > 25 | 98 | 26 | 1.0 | 0.38 (0.16, 0.90) | |
| Lung damage on computed tomography | | | | | 0.3173 |
| ≤ 50% | 75 | 6 | 1.0 | 1.05 (0.19, 5.81) | |
| > 50% | 125 | 45 | 1.0 | 0.32 (0.16, 0.68) | |
| Sequential organ failure assessment | | | | | 0.2718 |
| ≤ 4 | 132 | 17 | 1.0 | 0.46 (0.17, 1.26) | |
| > 4 | 67 | 34 | 1.0 | 0.26 (0.11, 0.61) | |
| Corticosteroids | | | | | - |
| Yes | 194 | 50 | 1.0 | 0.33 (0.18, 0.63) | |
| Colchicine | | | | | 0.5684 |
| No | 148 | 33 | 1.0 | 0.29 (0.14, 0.63) | |
| Yes | 52 | 18 | 1.0 | 0.51 (0.17, 1.54) | |
| Tocilizumab | | | | | 0.5710 |
| No | 174 | 39 | 1.0 | 0.31 (0.15, 0.65) | |
| Yes | 26 | 12 | 1.0 | 0.55 (0.14, 2.08) | |
| Renal replacement therapy | | | | | - |
| No | 185 | 38 | 1.0 | 0.44 (0.22, 0.88) | |
| Yes | 15 | 13 | 1.0 | 0.14 (0.02, 1.11) | |
| Sepsis | | | | | 0.8296 |
| No | 57 | 7 | 1.0 | 0.26 (0.05, 1.44) | |
| Yes | 143 | 44 | 1.0 | 0.35 (0.17, 0.70) | |
| Septic shock | | | | | 0.7716 |
| No | 142 | 17 | 1.0 | 0.30 (0.11, 0.87) | |
| Yes | 58 | 34 | 1.0 | 0.41 (0.19, 0.92) | |
| Acute kidney failure | | | | | 0.1220 |
| No | 171 | 30 | 1.0 | 0.50 (0.24, 1.06) | |
| Yes | 29 | 21 | 1.0 | 0.18 (0.04, 0.79) | |
| Arrhythmia | | | | | - |
| No | 186 | 42 | 1.0 | 0.22 (0.11, 0.47) | |
| Yes | 13 | 9 | 1.0 | 2.56 (0.62, 10.55) | |
| Pneumonia associated with IMV | | | | | 0.8398 |
| No | 156 | 21 | 1.0 | 0.33 (0.12, 0.89) | |
| Yes | 44 | 30 | 1.0 | 0.32 (0.14, 0.73) | |
| Catheter-associated bacteremia | | | | | - |
| No | 189 | 46 | 1.0 | 0.35 (0.18, 0.67) | |
| Yes | 11 | 5 | 1.0 | 0.14 (0.02, 1.35) | |
| Plateau pressure 24 h after IMV (cmH$_2$O) | | | | | 0.5694 |
| ≤ 30 | 147 | 27 | 1.0 | 0.30 (0.13, 0.69) | |
| > 30 | 53 | 24 | 1.0 | 0.45 (0.17, 1.20) | |
| Driving pressure 24 h after IMV (cmH$_2$O) | | | | | 0.8875 |
| ≤ 15 | 79 | 16 | 1.0 | 0.28 (0.09, 0.91) | |

(*Continued*)

**Table 3.** (Continued)

| Variable | Total | Death | HR (95% CI) | | P for interaction |
|---|---|---|---|---|---|
| | | | Low ($< 11.80$) | High ($\geq 11.80$) | |
| $> 15$ | 121 | 35 | 1.0 | 0.32 (0.15, 0.70) | |

HR, hazard ratio, CI; confidence interval; IMV, invasive mechanical ventilation.

pneumonia associated with IMV, plateau pressure 24 h after IMV, driving pressure 24 h after IMV.

We also used MI to maximize statistical power and remove bias. The MI was based on five replications and the Markov chain Monte Carlo method in the MI procedure in R to account for missing data on C-reactive protein and Procalcitonin. Results were similar to those of the initial cohort adjusted for potential confounders (Table 4).

## Discussion

In this study, the median duration of mechanical ventilation was 10 days, the median PEEP was 12 cmH$_2$O, and the mortality rate was 25%, which was consistent with previous studies

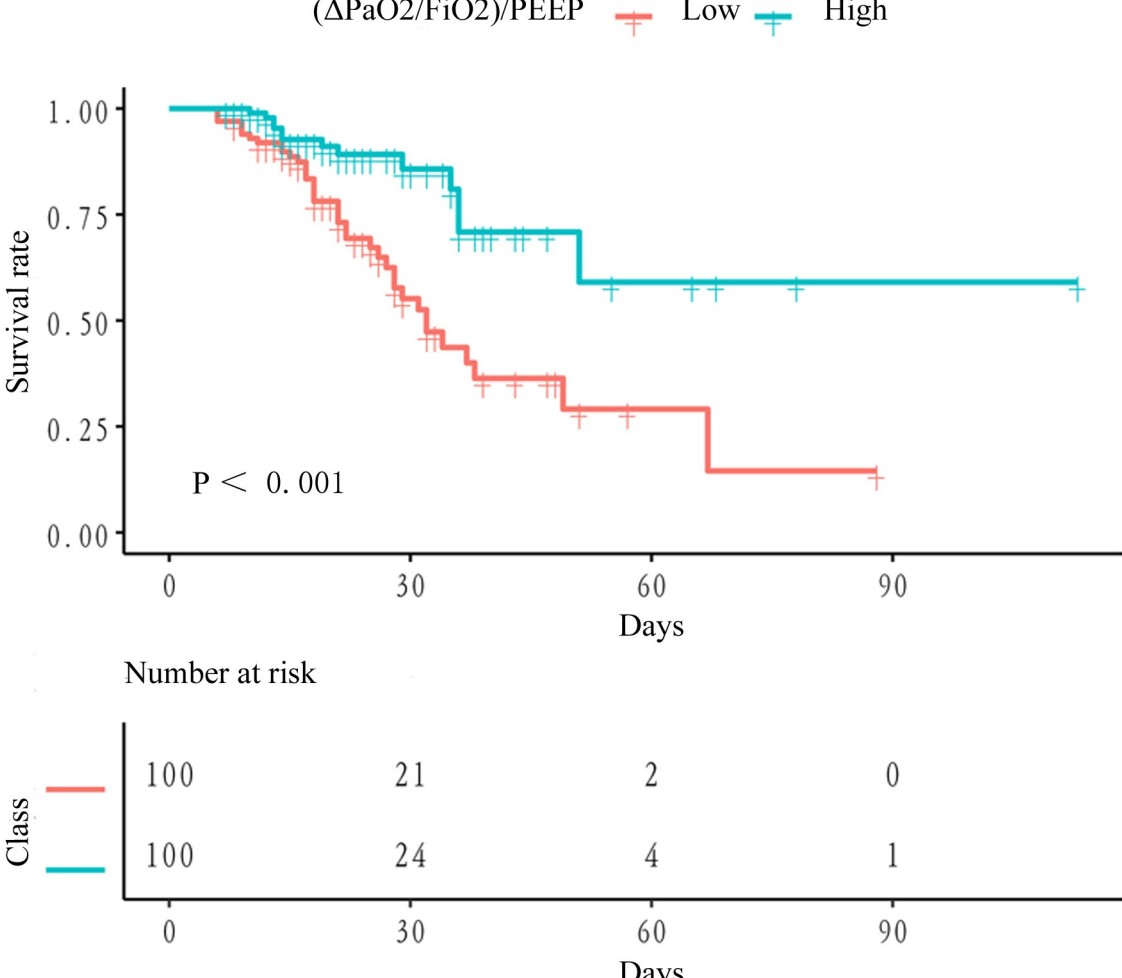

**Fig 1. Kaplan-Meier curves for patients in different ($\Delta PaO_2/FiO_2$)/PEEP groups.**

**Table 4. Cox regression model for In-hospital mortality.**

| Variable | | Unadjusted | | Model I | | Model II | |
|---|---|---|---|---|---|---|---|
| | | HR(95% CI) | p-value | HR(95% CI) | p-value | HR(95% CI) | p-value |
| Initial cohort | Low (< 11.80) | 1.0 | | 1.0 | | 1.0 | |
| | High (≥ 11.80) | 0.33 (0.17, 0.61) | <0.001 | 0.08 (0.02, 0.31) | <0.001 | 0.32 (0.11, 0.94) | 0.038 |
| | Per (ΔPaO₂/FiO₂)/PEEP | 0.90 (0.85, 0.95) | <0.001 | 0.82 (0.74, 0.91) | <0.001 | 0.88 (0.80, 0.97) | 0.013 |
| After MI | Low (< 11.80) | 1.0 | | 1.0 | | 1.0 | |
| | High (≥ 11.80) | - | - | 0.25 (0.11, 0.59) | 0.002 | 0.36 (0.16, 0.80) | 0.013 |
| | Per (ΔPaO₂/FiO₂)/PEEP | - | - | 0.89 (0.83, 0.96) | 0.002 | 0.90 (0.83, 0.97) | 0.009 |

HR, hazard ratio; CI, confidence interval; MI, multiple imputation.

[14,15]. Hypertension and obesity were the most common comorbidities in patients with COVID-19 pneumonia, and the majority of patients were male, which was also consistent with previous studies [16].

The $PaO_2/FiO_2$ ratio had some limitations. In patients requiring IMV, $PaO_2/FiO_2$ values could be overestimated on admission to the hospital or ICU [7]. On the other hand, $PaO_2/$

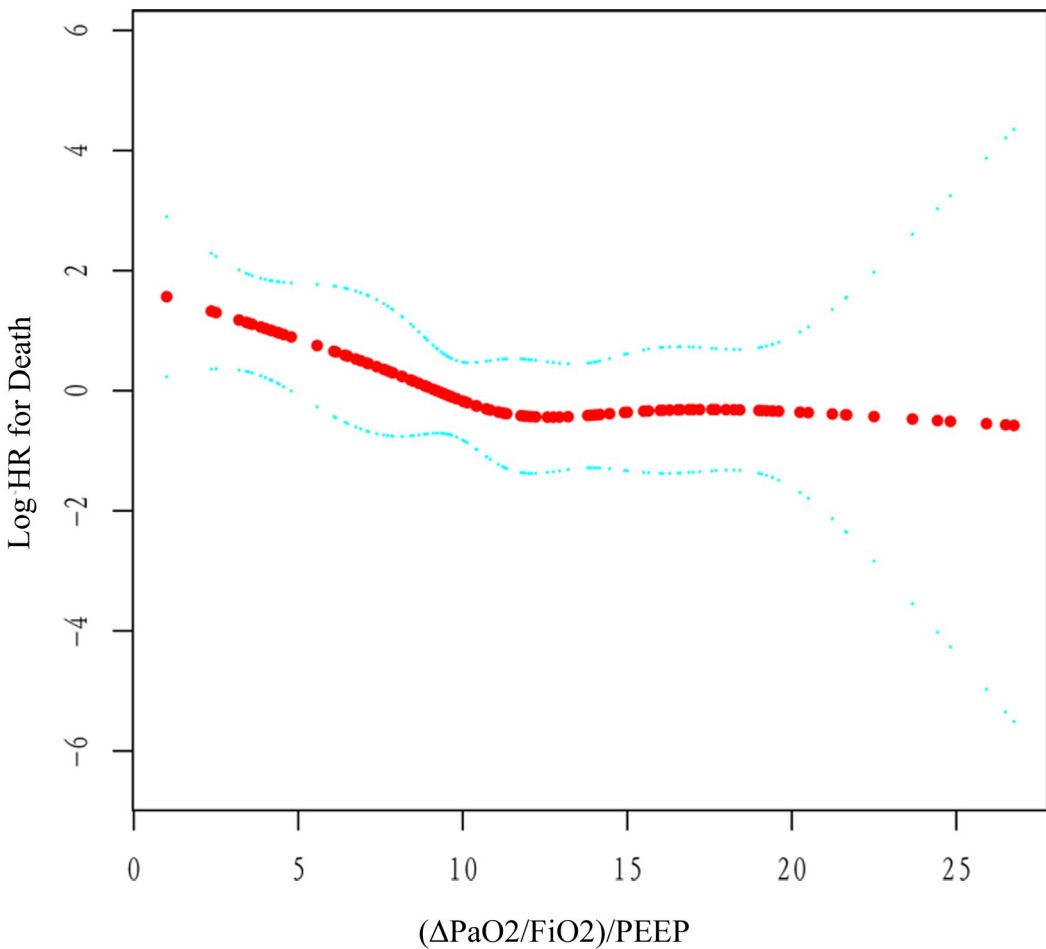

**Fig 2. Smoothed curve between (ΔPaO₂/FiO₂)/PEEP and in-hospital mortality.**

$FiO_2$ values before IMV might be underestimated in emergency situations such as sputum occlusion and respiratory arrest [17]. In addition, the mode of oxygen administration before tracheal intubation, such as nasal cannula oxygenation, mask oxygenation, high-flow oxygen therapy, and noninvasive ventilator-assisted ventilations, might also affect the pre-mechanical ventilation $PaO_2/FiO_2$ values [18].

Patients with significant changes in the difference in $PaO_2/FiO_2$ before and after intubation ($\Delta PaO_2/FiO_2$) might not have diffuse alveolar damage [19]. Patients requiring higher PEEP to prevent recurrent alveolar collapse might have combined more severe lung damage [20]. In this study, we found that the low ($\Delta PaO_2/FiO_2$)/PEEP group had lower $\Delta PaO_2/FiO_2$ values and higher PEEP values compared to the high ($\Delta PaO_2/FiO_2$)/PEEP group. After adjusting for potential confounders, the high ($\Delta PaO_2/FiO_2$)/PEEP group was associated with a 68% lower risk of death. Previous studies showed that a high percentage of COVID-19 patients admitted to ICU developed cardiac systolic and diastolic dysfunction [21–23]. Although higher PEEP values helped to improve oxygenation, high PEEP values could also lead to impaired right ventricular and hemodynamic function [24]. Patients might require additional fluid and vasopressors. Since higher values of PEEP could affect cardiac function [25], this might partly explain the higher mortality in patients with low ($\Delta PaO_2/FiO_2$)/PEEP group. Due to the limitations of the secondary study, data on cardiac insufficiency was not available in the primary dataset. We were unable to further analyze the effect of ($\Delta PaO_2/FiO_2$)/PEEP on the prognosis of patients with cardiac insufficiency.

In this study, all patients were placed in the prone position for 48 to 72 continuous hours [10]. To achieve the prone ventilation goal, patients might require high regimens of sedatives, analgesics, and neuromuscular blocking agents. Deep sedation in COVID-19 patients might be associated with several complications such as ventilation-associated pneumonia, prolonged MV duration, and ICU-acquired weakness [26]. These complications might further lead to prolonged ICU and hospitalization and increased mortality [26,27]. However, we were unable to obtain data on analgesic sedatives and neuromuscular blocking agents and therefore could not further analyze their effects on mortality.

## Limitations

This study had some limitations. First, this was a single-center retrospective cohort study to evaluate the association between ($\Delta PaO_2/FiO_2$)/PEEP and in-hospital mortality in COVID-19 patients requiring IMV. Thus, the findings might not be generalizable to the general population. Second, this study was a secondary analysis and could not further assess the effect of $FiO_2$ on ($\Delta PaO_2/FiO_2$). Third, due to limited resources, this study did not adopt an extracorporeal circulation membrane strategy to manage hypoxemia. Fourth, although we adjusted for possible confounding variables, other unmeasured variables might affect the results [28].

## Conclusions

The ($\Delta PaO_2/FiO_2$)/PEEP ratio was associated with in-hospital mortality in patients with COVID-19 pneumonia. ($\Delta PaO_2/FiO_2$)/PEEP might be a marker of disease severity in COVID-19 patients.

## Supporting information

**S1 Table. The regression coefficient in the basic model and full model.** IMV, invasive mechanical ventilation. These covariates produced over 10% change in the regression coefficient of ($\Delta PaO_2/FiO_2$)/PEEP and were adjusted in multivariate analysis when added to the

basic model or removed from the full model.
(DOCX)

**S2 Table. Stratified analysis between (ΔPaO2/FiO2)/PEEP and in-hospital mortality.** HR, hazard ratio; CI, confidence interval.
(DOCX)

## Author Contributions

**Data curation:** Youli Chen, Jinhuang Lin, Zhiwei Su, Tianlai Lin.

**Investigation:** Huangen Li, Zhiwei Su.

**Methodology:** Huangen Li.

**Software:** Youli Chen, Huangen Li, Jinhuang Lin, Zhiwei Su, Tianlai Lin.

**Writing – original draft:** Youli Chen.

**Writing – review & editing:** Tianlai Lin.

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
