## [Decision Letter · Decision Letter 0]

17 Mar 2024

PONE-D-24-03752Association between (ΔPaO2/FiO2)/PEEP and in-hospital mortality in patients with COVID-19 pneumonia: A secondary analysis

Short Title: (ΔPaO2/FiO2)/PEEP and in-hospital mortalityPLOS ONE

Dear Dr. Lin,

Thank you for submitting your manuscript to PLOS ONE. After careful consideration, we feel that it has merit but does not fully meet PLOS ONE’s publication criteria as it currently stands. Therefore, we invite you to submit a revised version of the manuscript that addresses the points raised during the review process.

**ACADEMIC EDITOR: **<ul><li>Thank you for your recent submission; your research paper has now been evaluated by two experts, and we have discovered significant information that can contribute to the scientific community. However, there are some issues that need to be corrected, so that we can reconsider your work further.

Please take the time to go over the reviewers' opinions, as well as their recommendations. Please respond to the points that have been brought up and handle the paper in the manner that has been advised.

Also please take care that

@ The aforementioned statistical points must be rectified and corrected.

@ Given the limitations mentioned, the conclusion should be rewritten in both abstract and at the end of the paper, also, mention that further randomized, well-designed trials should be conducted to investigate the findings in order to make generalized recommendations.==============================

We look forward to receiving your revised manuscript.

Kind regards,

Wael Hafez

Academic Editor

PLOS ONE

2. Please amend either the title on the online submission form (via Edit Submission) or the title in the manuscript so that they are identical.

Additional Editor Comments:

Dear Authors:

Thank you for your recent submission.

Your research paper has now been evaluated by two experts, and we have discovered significant information that can contribute to the scientific community. However, there are some issues that need to be corrected so that we can reconsider your work further.

Please take the time to go over the reviewers' opinions as well as their recommendations. Please respond to the points that have been brought up and handle the paper in the manner that has been advised.

Also, please take care that .

The aforementioned statistical points must be rectified and corrected.

Given the limitations mentioned, the conclusion should be rewritten in both the abstract and at the end of the paper. It should also mention that further randomized, well-designed trials should be conducted to investigate the findings in order to make generalized recommendations.

Reviewers' comments:

Reviewer's Responses to Questions

**Comments to the Author**

1. Is the manuscript technically sound, and do the data support the conclusions?

Reviewer #1: Yes

Reviewer #2: Yes

2. Has the statistical analysis been performed appropriately and rigorously? 

Reviewer #1: Yes

Reviewer #2: Yes

3. Have the authors made all data underlying the findings in their manuscript fully available?

Reviewer #1: Yes

Reviewer #2: Yes

4. Is the manuscript presented in an intelligible fashion and written in standard English?

Reviewer #1: Yes

Reviewer #2: Yes

5. Review Comments to the Author

Reviewer #1: (ΔPaO2/FiO2)/PEEP seems to be a good predictor of outcome in patients with respiratory failure secondary to Covid-19. However, it is likely be a marker of disease severity rather than a modifiable risk factor. It is not clear from the manuscript what suggestions the authors have for clinicinas at bedside for modifying this risk factor. Was there any evidence in the data to suggest that the PEEP or FiO2 was not titrated appropriately?

Following gramattical/typographical errors for the consideration of the authors

Line 167, Most commom complications should probably read as most common comorbidities

Line 169 - 172, could be reformated to draw the attention of the reader to the limitations of PaO2/FiO2 ratio

Line 193 insufficiency are not known could read as insufficiency was not available in the primary dataset

Reviewer #2: I have gone through the study and have found some merit. But there are some suggestions and points which must be accommodated before the final acceptance of study.

1. The authors have presented Hazard Ratio in the results section of abstract with their confidence intervals but its missing p-value. p-value increases the worth of results. For each HR the associated p-value must be mentioned.

2.The conclusion section of abstract is unclear. What is meant by linearly associated? On what grounds the authors declared it linearly associated. It will be better if conclusion is general not specific about relationship.

3.In line 117,"Their mean age was 54.29 years, and 42 patients (21.0%) were female". This line should be separated into two lines explaining about mean age with SD. The numbers and associated percentages should be mentioned for both male and female.

4. From line 117-125, the authors discussed the significant difference but did not mention the p-values or any result with the statement supporting their outcomes. It will better if significant or non-significant results be mentioned with each statement

5.In whole study, authors some time used past tense and sometimes present tense. It should be adjusted for the complete study and should have same pattern for complete study. In discussion, authors used present tense while after few lines past tense were used. It should be checked properly.

6.The limitation of the study should be separated from the discussion section.

7. The conclusion section is short. it should be expanded and should discuss other aspects of study also like theoretical and practical implications.

8. In table 1, p-value should accompany the test-statistic value also.

9.The authors mentioned in their study that Continuous data were presented as mean ± standard deviation (SD) if normally distributed. Which variables were tested normal and which were non-normal. If all the variables were non-normal then this statement should be excluded. it would create confusion between normality and non-normality issue. the statement can be "Continuous data were presented as mean ± standard deviation (SD)".

6. PLOS authors have the option to publish the peer review history of their article (what does this mean?). If published, this will include your full peer review and any attached files.

Reviewer #1: **Yes: **Kesava Ananth Ramakrishnan

Reviewer #2: No

---

## [Author Response · Author response to Decision Letter 0]

26 Apr 2024

Thank you for the editor and reviewers’ comments concerning our manuscript entitled “Association between (ΔPaO2/FiO2)/PEEP and in-hospital mortality in patients with COVID-19 pneumonia: A secondary analysis”. Those comments are all valuable and helpful for revising and improving our paper, as well as the important guiding significant to our research. We have studied comments carefully and have made corrections which we hope meet with approval.

---

## [Decision Letter · Decision Letter 1]

14 May 2024

Association between (ΔPaO2/FiO2)/PEEP and in-hospital mortality in patients with COVID-19 pneumonia: A secondary analysis

PONE-D-24-03752R1

Dear Dr. Lin,

We’re pleased to inform you that your manuscript has been judged scientifically suitable for publication and will be formally accepted for publication once it meets all outstanding technical requirements.

Kind regards,

Wael Hafez

Academic Editor

PLOS ONE

Additional Editor Comments (optional):

Dear Authors

good day

In light of your success in managing the paper as advised, we decided to approve the work based on the editor's and reviewers' comments, and it is now accepted. Congratulations!

Reviewers' comments:

Reviewer's Responses to Questions

**Comments to the Author**

1. If the authors have adequately addressed your comments raised in a previous round of review and you feel that this manuscript is now acceptable for publication, you may indicate that here to bypass the “Comments to the Author” section, enter your conflict of interest statement in the “Confidential to Editor” section, and submit your "Accept" recommendation.

Reviewer #2: All comments have been addressed

2. Is the manuscript technically sound, and do the data support the conclusions?

Reviewer #2: Yes

3. Has the statistical analysis been performed appropriately and rigorously? 

Reviewer #2: Yes

4. Have the authors made all data underlying the findings in their manuscript fully available?

Reviewer #2: Yes

5. Is the manuscript presented in an intelligible fashion and written in standard English?

Reviewer #2: Yes

6. Review Comments to the Author

Reviewer #2: The authors have accommodated all the suggestions. have improved the analysis section and followed directions to improve the study.

7. PLOS authors have the option to publish the peer review history of their article (what does this mean?). If published, this will include your full peer review and any attached files.

Reviewer #2: No

---

## [Editor Report · Acceptance letter]

22 May 2024

PONE-D-24-03752R1 

PLOS ONE

Dear Dr. Lin, 

I'm pleased to inform you that your manuscript has been deemed suitable for publication in PLOS ONE. Congratulations! Your manuscript is now being handed over to our production team.

Kind regards, 

on behalf of

Prof Dr. Wael Hafez 

Academic Editor

PLOS ONE